# Knowledge Breakthroughs through Time in Mind and Action—An Outline of a New-Old Knowledge History

Arne Jarrick

Department of Archaeology and Classical Studies, Stockholm University, SE-106 91 Stockholm, Sweden; arne.jarrick@arklab.su.se

**Abstract:** An outline of a research program on knowledge progress and strategies for knowledge breakthroughs through time, in two parts: A. An outline of five steps/stages in such processes and studies via a set of cases: 1. the inception of a new finding/discovery; 2. cognitive resistance to the new finding; 3. cognitive acceptance of the new finding but continuing resistance to its practical implementation or behavioural adjustment to it; 4. practical as well as cognitive acceptance; 5. openness (or not) to related further findings. B. Defense of my approach in view of criticism from representatives of new lines of research in what has been called knowledge history.

**Keywords:** knowledge; knowledge progress; knowledge history; knowledge strategies





## 1. Introduction

The human understanding when it has once adopted an opinion draws all things else to support and agree with it. And though there be a greater number and weight of instances to be found on the other side, yet these it either neglects and despises, or else by some distinction sets aside and rejects, in order that by this great and pernicious predetermination the authority of its former conclusion may remain inviolate.

[—]

The human understanding is not composed of dry light, but is subject to influence from the will and the emotions, a fact that creates fanciful knowledge; man prefers to believe what he wants to be true.

(Francis Bacon 1620/1960) [1].

The frequently quoted aphorisms above (the first more than the second) could stand as portal sentences for a lot of recently newly started and badly needed international research programs on knowledge resistance[1] [2,3], for example the one run by the Swedish philosopher Åsa Wikforss [4]. Like Bacon's lamentations about the state of mind of his fellow travellers in early modern Europe, this program focuses on the similarly "poor" minds of today's citizens, despite the fact that they are far more educated and enlightened than their ancestors normally were.

> "Taken together, these observations give a rather gloomy account of the history of humankind. Will we ever learn?"

However, the quote below may serve to temper such misanthropic pessimism. It is taken from David Wootton's summary of the cognitive worldview of a well-educated European at the beginning of the seventeenth century:

> "He believes that witches can summon up storms that sink ships at sea [...] He believes in werewolves [...] He believes that mice are spontaneously generated in piles of straw [...] He believes that a murdered body will bleed in the presence of the murderer [...] He believes that it is possible to turn base metal into gold [...]."

([5] pp. 6–7)

> "Today, almost no one, whether well-educated or not, believes anything of this. But already in the eighteenth-century a reasonably enlightened citizen of England had dismissed of most of these myths."
>
> ([5] p. 10)

Although it is likely that these enlightened citizens represented no more than five percent of the population, their attitude represented something new as well as something that in due course was to be broadly accepted.

Thus, despite all kinds of excuses for ignoring or efforts to resist knowledge, manifest in all generations, people have managed to adopt a lot of essential new knowledge which, furthermore, have become integral parts of their cognitive mindsets, often guiding them in daily life. Originally alien and controversial knowledge has managed to break through initially powerful resistance—time after time. So, what seems synchronously hopeless may seem less so diachronically, i.e., from a backward-looking and long-term historical perspective[2] [5]. Therefore, when studying knowledge resistance, one should also study the historical process where it becomes continuously and gradually overcome, not to mention some notorious and fundamental discoveries which never provoked very much controversy ([5] pp. 246–247). This is important, not least to give clues as how to find and apply effective cures against currently raging indifference or resistance—whether growing or not. This is the point of departure for my suggestion of a research program focused on the positive side of the history of human knowledge. Below, I will go into details about how it could be designed.

## 2. Aims

The program would specifically focus on knowledge gain where implications for human action can be identified. Ideally, the type of knowledge gains I am looking for is such that once adopted it encourages or even forces people in common (or the citizens/the subjects/the public and their leaders) to adjust their action accordingly. The aim would be to explain how this process works, i.e., to come to grips with the mechanisms behind the process where new knowledge, originally advanced and represented by a tiny minority, eventually not only becomes incorporated by a majority, or at least by a considerable number of people outside the knowledge producing sector itself, but also turns into a certain practice conditioned and driven by the new knowledge.

What I envision is a comparative study of cases of new knowledge that have been strongly questioned by considerable parts of the population, in relation to cases of new knowledge that developed without facing noticeable protest. More precisely: the task is to study both such cases where various interests to contest or resist new knowledge and its behavioral implications can be identified, and cases where such interests cannot be identified and where it is likely that they are absent. Such interests may have been of different types: intra- or extra-scientific, buried in institutional inertia, ("we have always done so and will go on doing so for ever"), commercial, ideological, et cetera. This could be one way of identifying causes and mechanisms. In addition, also some cases of more or less spectacular failures should be involved in the study, both those that were resisted by the contemporaries and those that were accepted. One example of the former, could be Galileo's false theory of the tides, an example of the latter some of Aristotle's many weird ideas.

## 3. Five Stages of Knowledge Progress

In this study, the destiny of knowledge would be followed in five steps, as if it progressed in five stages although in real life such stages are not discrete. This analytic procedure is built on the assumption that new knowledge must break through both cognitive and behavioral resistance before it is reasonable to conclude that it has been fully incorporated among its intended recipients or appropriators. In each of these steps or stages, key players/strategic proponents should be identified as well as the specific means they deploy to reach the addressees ([6] pp. 100; 328–330). Further down, I will try to identify possible

strategies and to map what have I have come across in this regard (section "Preliminary thoughts", pp. 9–13). Below, the five stages are distinguished the following way.

The process starts with the knowledge gain/breakthrough as such, i.e., with a discovery or a finding. This is stage one. In a second stage these findings are met with cognitive as well as behavioral resistance (or sometimes with non-resistance), to the new finding. Then, in stage three, comes the cognitive breakthrough, i.e., the point where a substantial fraction of the population—not yet specified or quantified—recognizes certain new knowledge as valid, without, though, yet adjusting their action accordingly—be such "inertia" either conscious and deliberate or less than conscious[3] [7]. At this stage it is not the knowledge resistance as such that blocks the process, but other mechanisms, not to be addressed here. Hypothetically, there are two kinds of circumstances conditioning behavioral changes driven by attitudinal changes, and one kind of circumstance where, conversely, it is a gradual change in habits that "silently" or semi-consciously is the driver of a change in attitudes/mentalities. These changes represent a fourth stage in the process. On one hand people might change their basic attitudes when they are facing reliable threats that would cause them unacceptable damage if neglected or denied, for example, a series of extremely hot summers or the COVID-19 pandemic. On the other hand, trustworthy predictions or "promises" of substantial improvement of people's lives, provided they change their behavior or habits might also bring about profound attitudinal shifts. Practical breakthroughs might also happen when cognitively incorporated knowledge gets markedly transformed into action among the population ([7] p. 110). Subsequently, new and more informed lines of action may generate positive feedback loops in the form of a widened openness to the acquisition of even more new and earlier unwelcome knowledge—and so forth. This would represent a fifth stage in the trajectory of knowledge progress, but it could also be seen as the beginning of a new cycle.

## 4. Analytical Problems to Consider in Relation to the State of the Art

The history of science, history of knowledge and intellectual history are certainly no new areas of research. So, to make sure that new knowledge about the long-term trajectory knowledge may be gained through my approach, I will here address the state of the art in relation to some of the most relevant analytical problems raised by contemporary scholars of these areas.

In contrast to earlier science historians, contemporary knowledge historians[4] [8] question whether it is meaningful or even possible to identify the specific intellectual and geographical space where a certain piece of knowledge was produced for the first time by a certain ingenious individual—or by certain individuals, for that matter. For example, the Swiss scholars Philipp Sarasin and Andreas Kilcher question the traditional historical preoccupation with origins and novelty. They claim that it is impossible to identify fixed origins for various forms of knowledge[5] [9]. Since the development of knowledge is a dynamic process, no such point of origin can be identified, they state. A step forward at one point in time is never the first step taken, but only one step on an endless pathway. But even if it were possible to identify a first step, today's knowledge historians consider such pre-occupation rather irrelevant since it is said to disregard the fact that research is always a form of communication (see below p. 6).

I would say, that this is the creed of the day[6] [9,10]—and that it is basically wrong. First, it is to deny the existence of a certain entity only because it depends on other entities for its appearance, as if one could never tell when a child was born just because it has parents and one day will become a parent itself. Thus, whatever the background of a certain discovery or finding or ground-breaking theory, it cannot but happen at a certain time and place, or in few places such as the neo-classical revolution in economics [5,11]. Of course, almost all discoveries are linked to other discoveries in the past and the future, which, however, does not make them less possible to pin-point both spatially and temporally. And it is their fate from such a spatio-temporal point of view that I am particularly interested in. If such an approach were not legitimate, then no one would be—except the one where

the Big Bang was taken as the point of departure [12]. Second, I find such an endeavor deeply meaningful, mainly because it can contribute to an understanding of one of the basic mechanisms behind cultural evolution, namely the process whereby intellectual conventions are transcended.

Nothing of this means that all discoveries follow the same path. Whereas some discoveries are even prospective (Halley's prediction that the comet would return in 1758), and others evolve over time (the discovery and understanding of oxygen), still others do virtually happen all of a sudden, such as Galileo's discovery of the moons of Jupiter and Archimedes' 'Eureka' [5].

As a corollary to the first point, knowledge historians stress that knowledge does not simply "trickle down" from its producers to its "consumers". Or as seen from the other side: people do not simply "receive" knowledge passively from above, but appropriate it in an active way. They are not just like empty vessels being filled with information. I think that this is a generally valid observation, pointed out by many scholars over the years [13,14], and it should be kept in mind when studying the spread of knowledge. At the same time, however, it should not be taken as a given. It is possible, even likely, that the incorporation of knowledge rather differs along a gradient from passive to active.

Now, the fact that knowledge normally is appropriated and not only consumed, means that the dissemination process is often less than straightforward, although far from always. It happens that the original cognitive message is transformed during its passage from the sender to the recipient—and subsequent recipients. Generally, the more it spreads, the more blurred and contaminated it may become, with external elements. This could mean that the original message gets lost at the end of the day. It may be caused by emotionally conditioned knowledge resistance, but it can also be due to plain cognitive difficulties in digesting complicated messages.

It should be noted in passing, that in this sense cultural transmission, of which knowledge is just one "trait", is distinct from genetic transmission where the vertical copying is normally faithful (apart from the mutations). However, for the new knowledge historians it is somewhat double-edged to recognize that human information may become corrupted in the dissemination process. On one hand, it would be to validate their criticism of the conventional trickle-down perspective, but on the other hand it may give the annoying impression that there is an unbridgeable intellectual gap between the circle of smart scientists and a less-well-equipped population. But, first, it cannot be denied that such bending towards misrepresentation does take place, however far from always. The debate in the 1920s and 1930s about Einstein's theory of relativity could serve as a case in point [15]. Secondly, when scientific findings are misrepresented along the road it is normally not a matter of difference in intelligence between experts and the public, but of difference in training. Sometimes, it might also be due to a difference in the strength of religious conviction—in some countries more than in others.

The rejection of the trickle-down approach is sometimes rather a moral or political critique of authoritarian ways of imposing knowledge on people, than a rejection of the observation as such that knowledge is sometimes vertically and faithfully spread. This should indeed be accounted for, given that one considers that the dissemination of knowledge is also a way to provide the citizens with empowering tools.

Another aspect of the dynamism of knowledge is that its dissemination is held to be circular rather than one-directional, i.e., rather a matter of communication than of dissemination, whether in a straightforward way or not [16]. Isaac Newton's work has been taken as a case in point. For the development of his theory of gravitation and everything related to it, Newton relied on observations made by a lot of ordinary people from many different places around the world. Hence, Principia Mathematica was not a one man's achievement, whose solitary discoveries radiated from its ingenious intellectual origin [17]. This frequently cited example is, however, not as clear-cut as it may seem at first sight. First, even if it were, one should not throw out the baby with the bathwater—the trickle-down perspective is not passé just because it does not fit all cases. Secondly, even in this case

certain vertical aspects of the process are obvious ([6] p.320). Consider the following: Newton's sources of inspiration and the world-wide reports he utilized are pieces of knowledge on a lower level of abstraction or generality than his theory of gravitation. They were building blocks for his synthesis, but the synthesis was his and it was far more than the sum of these contributions. Moreover, it was exactly this synthesis that triumphed in a world-wide trickle-down process, whereas the information he gathered travelled upward and inward to the center of his creative mind. Parenthetically, this applied to his theory of light and color, too [18]. And this is the case with a countless number of scientific innovations—whatever we might wish. As stated by Daniel Wootton concerning the development of water-wheels:

> "Those who think that modern science derives from the empirical experimental enquiries of craftsmen and artisans need to account for the extraordinarily slow evolution of water-wheel technology prior to the introduction of Smeaton's scientific method."
>
> ([5] p. 489)

There are a lot of other social contexts where such trickle-down processes are visible. It is particularly manifest in the school system. Here knowledge is imposed on youngsters whether they have asked for it or not, and the teachers try to make sure that the kids get it right, i.e., as it was meant to be taken in. The fact that knowledge here must be conveyed in a simplified form, does not contradict this—it is still a matter of trickle-down, although adjusted to different degrees of preparedness among the students. I am not sure whether knowledge historians have studied this arena of knowledge communication.

In my view, a closer look at some of the cases that the new brand of knowledge historians put forward in order to demonstrate that knowledge circulates shows rather the opposite: that it does trickle down. One such case is the discovery of the harmful cardiovascular effects of high levels of cholesterol, discussed in an essay by Finnish historian Laura Hollsten. She describes how this discovery, made in a medical lab, soon became widespread common knowledge, powerful enough to redirect people's behavior, in this case among the Finnish population [19]. That all this was questioned later on, does in no way change the original direction of this cognitive-practical chain of events, where, furthermore, a scientific point of departure can be easily pin-pointed.

One implication of the far-going and mystifying notion that knowledge has no identifiable origin—being communicated without being produced, as if enacting an orbit without beginning or end—is that the scientist has no privileged position in the knowledge process. This obfuscates the irrefutable fact that a whole lot of knowledge is attainable only after years of training, hard work and trial and errors. There is nothing special about this; it is basically similar to carpentry or playing the violin et cetera, which also require thousands of hours to master [20].

This means that the scientific advancement of knowledge, which indeed is distinguishable from its dissemination, in most cases—for sure, not in all cases—is, and must be an elite activity. Normally it is practiced by members of a certain profession, although this is not, and has not, always been the case. In early modern Europe, scientific activities were often pursued through informal networks, frequently called academies ([5] pp. 340–341, 561). Yet, these activities also required much training. To imagine that it happens without this is nothing but wishful thinking, even if it is based on the correct notion that researchers qua researchers interact with other parts of society as much as they do in their roles as citizens ([20] pp. 5, 10–11, etc.). Moreover, to deny this is to play down the profound importance of basic research[7] [1].

If it were true that knowledge had no origin, then it would also be futile to try and distinguish research from the communication of research. Denying such a distinction is precisely what knowledge historians do—it is even constitutive of their approach ([19] pp. 19–20, 22), ([21] p. 220), ([22] p. 272)); the mission is to eradicate "the distinction between the making and the communicating of knowledge", as expressed by James Secord

([10] pp. 661, 663, 667). But it is wrong again. To infer from all scientists' involvement in all kind of societal affairs, whatever their role there, that science cannot be distinguished as a separate occupation and activity, is no better than claiming that two individuals cannot be distinguished just because they are a couple and comprise an interacting entity (hopefully). Obviously, the laboratory is a specific place where specific knowledge generating things happen that do not happen elsewhere. The same applies to the archive where historians delve, and so forth. The adherents of the "circulationist" perspective do themselves convey such examples, countering their own message. This is as inconsistent as their denial of the possibility of distinguishing between the domain of knowledge advancement and the domain of its dissemination, while at the same time recommending researchers in the field to shift their focus from one domain to the other—as if they were distinguishable nonetheless ([9] pp. 19, 22–23, 26–27), ([23] p. 160.).

One example of a flawed attempt to rub out the borderline between the advancement and dissemination of knowledge is an article on the communication of statistical knowledge during the nineteenth century. Here it is stated that the public representations of statistics were inseparably interwoven with the development of the statistical science as such. This is true—but only partly so. First, it is made clear in the article that the producer of new statistical knowledge was one and the same person as the one who conveyed it to the public. Secondly, the author circumvents the obvious fact that the development of new statistical methods was an intra-scientific affair. It was not the 'people' who invented rxy, statistical significance or the confidence interval [24].

I would say that this far-fetched denial of the distinction between different steps of the knowledge process resembles the criticism of the 'linear' model put forward by representatives of the so-called triple helix model ([20] pp. 6, 11–12).

Here, it may be appropriate to consider Joseph Needham's nuanced discussion on the very long-term development of technology and civilization in China, which, although focusing on technical progress, is relevant for the development of knowledge in general[8] ([25] pp. 228–229). Summing up his almost all-encompassing study in an impressively simple way, Needham states that things that are difficult to grasp in the first place are discovered or invented only by a few individuals in very few places, after which they can become established on a large scale only through diffusion. The opposite applies to things that are easily realized: they are independently introduced in many places and may get established without diffusion. Lastly, the more advanced a culture becomes in cognitive terms, the greater its potential to autonomously find solutions to advanced problems, and the less will be its dependence on intellectual copying of inventions made elsewhere. However, the last observation might be questioned, due to the obvious fact that it is far easier to copy inventions in a globalized and electronically connected world, than it was in a distant past where small groups of people were isolated from each other [26].

These reflections could be translated to the knowledge domain in general, simply implying that some steps forward may materialize in a circular way, whereas other steps necessarily become enacted as a trickle-down process.

Knowledge historians stress that the exchange of new knowledge between Western nodes and other intellectual centers has never been a one-way affair. Instead, it has been a matter of mutual give and take, i.e., a truly circular process. Here, a lot of new and solid knowledge has been gained, proving older Euro-centric views largely invalid [27]. At the same time, however, this observation has too often been used to argue in defense of another less substantiated claim: that the exchange of knowledge between intellectual elites and the public has been as circular as the exchange between the West and the East or South[9]. Thus, they confuse one valid observation with another less valid one. One exponent of this is Catherine Jami, who in an essay on cartography describes the mutual knowledge encounters between France and China from the late seventeenth to the mid-eighteenth century ([28] pp. 53–57). At the same time, she shows, perhaps unintentionally, that knowledge that was transmitted back and forth between these nodes was the exclusive privilege of a tiny knowledge producing and knowledge appropriating minority. Another

example concerns the British cartography in India during the Mughal rule. Here Kapil Raj presents the same view as Jami. Like her, he fails to make a distinction between the obviously circular, although hardly free and open movement of knowledge, and the synthesizing of knowledge gathered by a minority or even by single individuals. In this case it was done by the British cartographer James Rendell (1742–1830) ([29] pp. 49–66).

Thus, it is important to distinguish between the geographical direction of knowledge and the question of whether it trickles down from a minority, wherever this minority happens to dwell. In a typical formulation, Lissa Roberts conflates these two processes: "The historical and historiographical equation of circulation with diffusion of western knowledge...to the rest of the world..." [30]. Again, "diffusion" is one thing, "of western" another. If these processes are conflated, it risks producing an ideological bias against the recognition of the fact that knowledge often spreads from a primary node—or from a few primary nodes.

In passing, knowledge historians often refer to the development of cartography, so often that it makes me suspicious. Is it a case of cherry picking? There are, though, other cases pointing in the same direction. One example of this is pharmacology, where a continuous exchange of theoretical findings and practical experiences were going on between natural historians and pharmacists in the pharmacies of the urban centers of early modern Europe ([31] pp. 44–60). The botanical collections of the sixteenth century also fit into this picture ([32] p. 89). Perhaps, this is the case with natural history in general [33]. And there are more examples of such horizontal or circular encounters, exhorting us to modify the conventional vertical view on the communication of knowledge. However, to modify is something other than totalizing the presently fashionable circulation perspective. It rather serves the need to recognize that there is great variation in the means by which knowledge is advanced and communicated—like in all human culture. It is a way to shun cherry picking, to rule out all attempts to circumvent counter instances. For example, in eighteenth-century Britain a certain procedure was applied, through which ordinary citizens were invited to aid the scientists to gather information, although, in a way that was designed and controlled by the scientists. Thus, it was not a matter of equal exchange or circulation of knowledge [34]. To a certain extent this resembles what goes in in ornithology today, where ordinary citizens record and report sightings of birds [26].

In an essay on the questionable universality of knowledge, Lissa Roberts sets out to challenge Western claims to truth of certain theories that have a Western origin—here, as if they had an origin after all. It is an outflow of certain well-known anti-scientific ideas that no knowledge can be anything but local[10] [35]. In doing this, she inserts the dubious little word "seem" at strategic points in the argument. As is so often noticed: the devil is in the details. Thus, she refers to Newton's theory as an example of theories that "seem [my italics, AJ] to hold not only in London, but in Beijing, Lima, and Dar es Salaam as well". She continues by stating that "we need...to pose the question of how one can trace the move from local to the *seemingly* [my italics, AJ] universal character of scientific...knowledge" ([30] p. 14). She concludes the essay in the following way:

> "We have...seen how each of these visions considers the *seemingly* [here it is sneaks in again, this deceptive little word, AJ] evident claim of science's universal validity—as a reflection of objective truth, a manifestation and tool of western hegemony..."
>
> ([30] p. 24)

One wonders what this really implies: that Newton is valid only in London, or, if invalid, invalid only there? Does not gravitation work on people outside the Western hemisphere? Or as expressed by Newton's associate Roger Cotes: "For who doubts, if gravity be the cause of the descent of a stone in Europe, but that it is also the cause of the same descent in America?" ([5] p. 322). Moreover, what happens with the idea of intellectual reciprocity between West and South if European theories are regarded as imposed on the rest of the world?

Some knowledge historians go as far as to deny that real knowledge is attainable at all. For them, knowledge is a game, where traditional science historians—like other traditionalists—pretend that they discover facts that they are rather inventing. A paradigmatic example of this approach is Steven Shapin's and Simon Schaffer's Leviathan and the Air-pump, frequently referred to by knowledge historians since its publication in the year 2011 ([5] pp. 41–47). One wonders of course how even such an observation, tending towards a claim to truth, can be substantiated. However, such a dated attitude should be taken seriously only in the sense that it is just one aspect of knowledge resistance.

An older approach to knowledge change, akin to this, is represented by Thomas Kuhn and his followers. There are indeed some connections between my basic idea and his, but there are also significant differences. Kuhn was exclusively devoted to the logic of intra-scientific breakthroughs. It may be the case that the extra-scientific breakthroughs follow the same logic, but this we do not know. However, this is precisely what is at stake here. As far as I know, he did not carry out any empirical investigations on the issue. He did not inquire into the practical implications of the paradigm shifts he studied.

Against the background of the above discussion, I still find it legitimate to proceed with the project plans presented at the beginning of this article.

## 5. Which Cases and Why?

How does one identify the most interesting and most relevant cases? Is it enough to carry out case studies by following a series of knowledge breakthroughs trough history? Or is it also necessary carry out experiments aimed at testing the gradient of people's varying readiness to adopt new and 'awkward' knowledge?

Anyway, by following the knowledge process over time, I hope to clarify whether it has gone through important changes in terms of its characteristics and its mechanisms. When selecting cases to investigate, there are some dimensions to consider, such as generalizability, types of cases and explanatory factors.

The cases taken together should maximize the possibilities of making general inferences, however from a repertoire of cases representing variation. Both unambiguous and ambiguous research findings should be picked, like knowledge trajectories where people's behavior either changes before or after the cognitive change. Furthermore, successful as well as unsuccessful knowledge gains should be studied, i.e., those that were and those that never were implemented. The same applies to knowledge gains that did or did not face resistance, and knowledge gains that got lost and were reclaimed (electric trams; food habits; the healthiness of reasonable exposure to bacteria; the magnet; etc.).

And what mechanisms, i.e., what potential explanatory factors, are to be searched for and found? Is it a matter of the magnitude of the consequences of knowledge breakthroughs among people in general, as well as the consequences of scientific knowledge progress that failed to break through among people in general? Has it something to do with knowledge breakthroughs of relevance for many people versus those of relevance only for few, as well as whether the practical breakthrough is perceived by people as something imposed from above, as a repressive exercise of power, or as "empowering" the citizens? What role is played by institutional conditions for the cognitive and the practical breakthrough of new knowledge? And so on. These are just some of the factors to be sought behind the different short- and long-term fates of new knowledge.

Needless to say, there are many possible candidate cases for empirical studies. This being so, ideally it will be possible to discern all the five stages in each of the cases I choose to study. One exemplary case would concern Marcello Malpighi and the capillaries. In the year 1661 Marcello Malpighi discovered the capillaries, the very thin system of vessels linking the arteries to the veins. It had been made possible thanks to the invention of the microscope [36,37]. The following has been said about the potential practical implications of his achievements in general: "He was vigorously denounced by his enemies, who failed to see how his many discoveries, such as the renal glomeruli, urinary tubules, dermal papillae, taste buds, and the glandular components of the liver, could possibly improve

medical practice." [38]. What has been the use of Malpighi's finding? Malpighi did not know what most people know today, that the capillaries are responsible for the exchange of gases, nutrients and waste products between the tissues and the blood. The discovery of the capillaries provoked much resistance among Malpighi's colleagues in Bologna and in 1684 his home was torn into pieces and his manuscripts thrown out and dispersed by his fierce opponents. To get away from all this hostility, he moved to Messina, invited by Giovanni Borelli, and in 1691 he was invited by the Pope in Rome to become his surgeon. So, the hostility against him was not as pronounced in other parts of Italy. He was already a member of the Royal Society already by 1667. What about people outside the circle of scholars? I have no clue—did anyone at that time?

There are of course innumerable other relevant cases, such as the Newtonian mechanics, the destiny of the discovery and of the function of the blood stream, Joseph Lister and the use of carbolic acid as prophylactic antiseptic treatment in surgery [39], and climate change, perhaps the most paradigmatic of cases, et cetera. The selection of cases remains to be decided.

## 6. Preliminary Findings

*The Repertoire of Strategies*

To achieve something, one has to study both the nature of knowledge resistance, or rather its "natures", and the strategies aimed at overcoming it, or them. Below I present my preliminary findings of the latter in terms of four such strategies. They are not entirely distinct, but represent four reasonably distinguishable pathways to the acceptance of knowledge breakthroughs. However, taken together they could be seen as a repertoire from which ingenious minds could draw on and combine freely. Sometimes one of them, or a combination of more than one, may have been considered more suitable than another—and vice versa. The question could still be asked as to whether one of these strategies is generally superior to the others, or if each could be successfully applied—separately or in concert—according to varying conditions for the reception of new knowledge. Parenthetically, it might be relevant also to go into some detail about the various knowledge discovery strategies applied, since the choice of such strategies might have had an impact on the reception of the new knowledge gained.

Scholars applying what I, for better or worse, choose to call the *low-key approach*, take a lot of steps trying to avoid provoking skeptical minds, because they anticipate resistance to their new findings. One measure taken is to downgrade the magnitude of the required intellectual change due to the new finding, presenting it as an extension of prevailing knowledge rather than as a revolutionary upheaval, even though it may be the latter. They could even present it rather as a re-discovery of old knowledge than a discovery of something new, or conceal the most challenging or provocative parts of the new finding.

They may also manage to make a minor rift in the wall of confidence vis-à-vis hailed intellectual heroes, such as Aristotle, by pointing out some of their most *insignificant* flaws instead of attacking their major errors. Likewise, it could be a good idea to hold on to traditional and established *forms* of knowledge, such as concepts, definitions et cetera, while modestly favoring the new *content*, i.e., a new way of seeing it in the wake of the new insights. By doing this they may contribute to a process whereby the new scientific content will gradually be perceived as natural or self-evident, to a point where the old forms of knowledge will be considered obsolete and become replaced by new forms. Eventually, the new "liquid" (the new scientific content) will make the old "pot" (the form) "corrode" from inside, and the need for new forms will be badly felt and no longer considered controversial. Thus, one should not push the formal change, but just let it happen spontaneously and when time is ripe.

Yet another ingredient in the low-key approach is to postpone the presentation of the new findings until the supposedly most "receptive" moment emerges, while in the meantime sending up pilot balloons. Such "opportunism" may be supplemented by

flattering the potentially most suspicious among the most powerful in society, as a means to pave the way for their receptiveness to the new findings.

One example of this rather cautious approach is John Maynard Keynes, who with his macro theory challenged conventional economic thinking (new content), while keeping its conceptual system (forms) [40].

An example from early modern Europe is Newton's theory of light and color as he presented it in his mature years: "Later, in the Opticks, he adopted a more accommodating approach and presented his discovery as an extension of existing knowledge." ([18] p. 67). Newton's Principia, published in 1687, was described as a defense against superstition (such as a belief in witchcraft and miracles) but also against atheism—materialism dressed up in a Christian outfit ([5] pp. 466, 471, 473–474).

A hundred years earlier, Copernicus did something similar. He referred to "Philolaus the Pythagorian (c. 470-385 BCE)" being at least "as an important precursor in proposing a moving earth [...]" "Even Galileo [...] repeatedly coupled Copernicus' name with that of Aristarchus of Samoa [...], to whom he (mistakenly) attributed the invention of heliocentrism"[11] [5,41,42]. Fearing ridicule, he also chose to publish in Latin, not in the vernacular ([41]).

Galileo, who already in the 1590s was convinced that Copernicus was right and Aristotle and Ptolemy wrong, also decided for the time being not to make his position publicly known ([41] pp. 50, 53). When Urban VII was elected Pope in 1622, Galileo wrote letters to him, his close friend in the past, in the most ridiculously subservient manner, praising him in a high-strung way and providing him with copies of all his books ([41] pp. 98, 128). He also sent up pilot balloons in favor of the Copernican system ([41] pp. 131–134). And due to warnings, in 1625 he postponed the completion of the *Dialogue Concerning the Two Chief World Systems* ([41] p. 141). Moreover, the dialogic form of the treatise was a calculated way to hide his basic views behind a fictitious figure ([41] pp. 146, 159).

As is well-known, Darwin deliberately postponed the publication of his theory. According to some, he hesitated because he feared that his findings could be politically misinterpreted or abused by radicals who, in the long run even might even threaten his own social position ([43] p. 20). Then, he was forced to let go when he received a paper by Alfred Wallace, where the same theory as he had been working on for decades was presented ([43] p. 131). But until then his attitude might be described as a low-key approach [44]. Moreover, in The Origin of the Species he said almost nothing about the implications of his theory for humans, [44–46] and his daughter helped him to leave out some sequences that they thought might repel the moral minds of their time too much ([46] pp. vii–viii).

So far I have identified two ingredients in what could be called an upfront approach aiming at as broad an acceptance as possible of new knowledge. Firstly, scholars applying such a strategy show no uncertainty about the validity and sustainability of the new results. Rather the opposite. They behave and express themselves is if they were completely certain about the correctness of their findings. Secondly, they seem to take steps to increase their courage by organizing groups of dedicated supporters, ([47] pp. 116–119) which indicates that they might be less than absolutely self-confident despite their upfront appearance.

Again, Newton's theory of light and color, as he presented it in the beginning, when he was in his thirties, is a case in point. As stated by Alan Shapiro:

"Thus, two reasons immediately present themselves to explain the difficulties Newton's theory initially encountered: its revolutionary nature and its superficial presentation [...] he was overconfident. He believed that his theory followed unambiguously from his experiments and that everyone would also immediately recognize this [...]"

([18] pp. 66–67)

And what about Galileo? Perhaps his self-confident attitude when he was young may serve as an illustration of an over-confident attitude. At this time he made fun of Aristotelians in a humorous provocative way, until he in 1616 was reproved by the Pope,

which made him silent on cosmic matters for seven years ([41] pp. 8, 50–51). His treatise *The Dialogue Concerning the Two Chief World Systems*, where he makes a fool of the pope Urban VII, transparently disguised as the stupid "Simplicio", is another manifestation of his irresistible propensity to tell the truth, ([48,49] p. 133), soon to condemn him to house arrest for the rest of his life.

A contemporary example could be the Swedish scientist Hans Rosling, who frequently appeared in mass media, stubbornly just telling journalists and others that he simply was right and that anyone opposing him was outright wrong. Period!

Two more strategies deserve to be briefly mentioned: one that I call the honest approach and the other which could be labelled the targeting approach.

It is characteristic of the researchers applying the former strategy, that they present their new findings as they are, without any rhetorical twists, and that they unhesitatingly argue in favor of formal adjustment if it is considered to be needed, ignoring possible resistance to it. Furthermore, they take a fallibilist attitude to their (and other's) research results, revealing actual uncertainties, though without shying away from showing confidence in solidly substantiated findings. Finally, it seems that many such scholars are ready to indiscriminately offer the new knowledge to all and anyone.

The honest approach might be just honest, but it could as well be a secret plan to overcome expected objections by being the first to address them. In his autobiography Darwin explains the almost immediate success of *The Origin of Species* by referring to his unusual habit of being as meticulous in considering counter-instances to his theory as observations which harmonized with it. Thanks to that, he anticipated many objections, which he responded to in advance, which is why he faced very few objections when his theory was made public [43].

The latter strategy is a way to economize with intellectual energy, by targeting those with the highest potentially positive sensitivity to the new findings, while ignoring or circumventing the dedicated conservatives, those who for the time being—they may be a whole "generation"—are in power. Perhaps Joseph Lister could be seen as an exponent of a targeting approach. In opposition to most of his older fellow surgeons, he introduced carbolic acid as a prophylactic antiseptic treatment in surgery. My impression is that he directed his efforts mainly toward younger surgeons who he considered relatively unprejudiced, finding it almost hopeless to try to change the minds of his older colleagues ([39] pp. 196–198).

This leads me to the reflections below.

It took about 30 years for William Harvey's discovery of the circulation of blood to be generally accepted among the circle of experts, ([36] p. 208) and the same goes for Joseph Lister's insights into the significance of fighting devastating bacteria with anti-septic means (carbolic acid) in surgery ([39] ch. 10–11, Epilogue). Isaac Newton's theory of color and light experienced roughly the same career. Indeed, from its inception at the beginning of the 1670s until a consensus on the truth of Newton's theory was reached, it took 50 years rather than 30. At last, in the 1720s Newton's theory had attained the stage of mainstream knowledge, and those few who still opposed it were no longer taken seriously ([18] p. 126.) When the Europeans first learned of the existence of America a new era started, which in 40 years' time lead to the recognition that new knowledge was indeed possible to attain [5,7][12].

These examples indicate a generational effect of scientific progress. If so, one strategy would simply be to patently wait until the older generation passes away, or to ignore it and direct all energy toward the new generation.

This relates to two other observations.

The history of knowledge abounds with innovative minds striving to gain support for their novelties from hesitating or even hostile contemporaries. Once they manage to gain general acceptance for their ideas/findings/discoveries/theories, it is likely that many of them put more and more effort into consolidating the newly won mainstream status of these ideas and less effort into "producing" new findings. Gradually, this may make them

less innovative and more conservative, at least if they take their achievement seriously. Probably, they want to prevent what they have accomplished from being ruined by the next generation. In due course, they will nevertheless be challenged by new generations of innovative individuals, who like them, their forerunners, aspire to establish their novelties for good. And so on, in a never-ending tug-of-war between innovation and tradition.

The tension discussed in the previous section can partly be explained by the well-known and well-established so-called confirmation bias. People tend to stick to early acquired ideas, which is why renewal often takes place only through the emergence of new generations.

Altogether, these reflections may serve as the rationale for the "targeting approach", but also for the low-key approach.

It is obvious from the tentative discussion above, that scholars have applied different approaches from time to time, in the way that both Newton and Galileo did. This may be due to their deliberate considerations of the particular appropriateness of a certain strategy at a certain time vis-à-vi certain people. However, it may also be due to more or less spontaneous ways of approaching people, sometimes possibly being an outcome of the age of the scholar. As a young man, Newton behaved more toughly than the elderly Newton who approached people in a more discreet way, fully aware of the resistance he could expect and how to overcome it.

At this stage of analysis this is of course pure guesswork, remaining to be substantiated. The point to be made is just that a study of such strategies has to consider not only the point of view of the sender but also how scholars may have conceived of the receptiveness of the addressee of the scientific message. Furthermore, how the sender acts, affects the recipient's reactions, and vice versa, in an ever-ongoing dynamic interplay between them, continuously changing the conditions for knowledge progress and its transformation into practice.

For example, in the sixteenth century something very similar to a heliocentric world-view had been around for about fifteen hundred years. Although being an undercurrent, it was rather well-known among learned people without stirring up very much intellectual unrest among them or among the authorities. However, this changed dramatically at the beginning of the modern era, when it increasingly became seen as a threat to the social and mental stability of the society, leading to a lot of trials and death penalties against the most outspoken adherents of the new-old world-view. Why? Well, as long as heliocentrism was still was an unproven idea, a hypothesis among other hypotheses, it could be seen as rather harmless. But, once more and more hard evidence was presented in its favor, it became increasingly threatening to those who held on to the old geocentric view while being stripped of most of their arguments for a *Weltanschaung* that they considered extremely important to uphold. This is the cunning of history, illustrating the dynamism of knowledge progress: the stronger the evidence for a certain finding, the more desperately resolute the resistance to it among many people. Evidence brings resistance, but, nonetheless eventually makes its way[13] [50]. How and by what mechanism, that's the issue at stake in project outlined above.

**Funding:** This research received no external funding.

**Institutional Review Board Statement:** Not applicable.

**Informed Consent Statement:** Not applicable.

**Conflicts of Interest:** The authors declare no conflict of interest.

## Notes

[1] Among those quoting it, see [2,3].

[2] For a sound defense of a backward-looking historiography, see ([5] ch. 16).

[3] A source of inspiration could be Daum, Andreas W. Varieties of Popular Science and the Transformation of Public Knowledge: Some Historical Reflections, Isis 2009, 100; pp. 328–330, although I am strongly critical to some of his ideas [6].

4    For the delayed decline of smoking as one example, see [8].

5    For an overview, see [9].

6    The most influential and most trend setting contribution is Secord, James A. Knowledge in Transit, Isis 2004, ([10] p. 95). Also see for example [8].

7    For an early and elegant defence of basic research, see Bacon, 2000/1620; [1] p. 81.

8    I am thankful to John McNeill for pointing out this to me.

9    For example, see [6]; pp. 324–326.

10    [5] pp. 513, 515, 528; [10] p. 659. For a critical remark, also see p. 660. Also [35]; ch. 30. But here it does concern the locally situated emergence of knowledge (as distinct from Newton's, Darwin's and other's claims), rather than its pretended local validity.

11    See: ([5] p. 78;) ([41]; p. 66.) Also see ([42] pp. 81–83).

12    [5]; p. 79. Also see Myrdal, 2008; p. 221, referring to Lee Smolin.

13    For a very illustrative example of this, see [50].

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
