# Peer review of "Knowledge Breakthroughs through Time in Mind and Action—An Outline of a New-Old Knowledge History"

_knowledge, doi:10.3390/knowledge3020011_

Round 1

Reviewer 1 Report

The English is a little rough here and there and there are some typos.
I particularly liked the ”repertoire of strategies” — it’s obviously rather preliminary and needs development, but it is thought-provoking.
For an extended discussion of resistance to new knowledge, see David Wootton, Bad Medicine (which should be read in the paperback edition which has an appendix not present in other editions).
It seems to me one might also develop a repertoire of knowledge-discovery strategies. We might distinguish bold innovators (Galileo, Newton) from conservative fixers (Clavius and the Jesuits, for example in their response to Galileo who acknowledge the validity of his key discoveries, but try to moderate their impact and originality) and both from team players (those engaging in what Kuhn called ”normal” science).
Bold innovators often attach themselves to an idea before it can be said to have been convincingly demonstrated — Galileo, for example, is a premature Copernican, who makes his discoveries precisely because he already believes Copernicus to be right though the evidence in support of this is very thin.  Bold innovators are sometimes playing the odds — the chances of success may be low, but the potential rewards high — or they may have a quite different way of ordering the tests that a successful theory should pass than their contemporaries — mathematicians, for example, in the Renaissance, are much less interested in authority and tradition than philosophers.
One needs to consider interesting failures: a striking example here is Galileo’s theory of the tides, which he thinks is the decisive evidence in favour of Coperncianism and which is simply false. It is remarkable a) that he was so heavily committed to it, despite obvious problems and b) that nobody else was convinced by it. This is a striking case of new false knowledge being successfully resisted. (see Wootton’s essay Galileo’s Failures in a volume called Causation and Modern Philosophy.
I hope these comments are helpful, either to develop this essay or in later work growing out of it.

Author Response

  1. The reviewer considers my English "a little rough". It may be so, and I would be ready to "refine" my English if she/he just gave som examples of the said roughness. However, the note is a little bit confusing, since the second reviewer considers my English "almost always correct". 
  2. The reviewer suggests that I might add a discussion  of various knowledge-discovery strategies, to distinguish bold ones from more conservative ones etc. This is a good point, which might be developed in another article. This is, however, not at the center of my interest here which is the success or failure of knowledge, whatever the knowledge-gaining strategies behind.
  3. The reviewer points out that also interesting failures should be considered, such as Galileo's theory of tides, luckily resisted (for good or bad reasons). This is a good point, to be developed either in this article or in another one.
  4.  The reviewer gives me two options, either to develop her/his comments in this article or to do it in later work. I understand this such that it would be acceptable tob do the latter.

Reviewer 2 Report

This is a well-informed article with some original ideas. In current studies of the history of knowledge, J’s emphases on breakthroughs, resistance, stages, strategies and the relation between knowledge and action are all unusual and worth developing, perhaps in several articles rather than just one. Some perceptive observations are made in passing, e.g. the idea of the ‘gradient’, p.4. I do have some criticisms, below, but the author should be able to respond to all of them in a second draft of the article.

1.    Surprisingly in view of the author’s career in general history and his past publications, the article is almost exclusively concerned with the history of science, rather than the history of knowledge in general. It makes unnecessarily frequent use of one book, David Wooton’s Invention of Science.

2.     In his critique of what he calls the ‘fashionable’ idea of the circulation of knowledge (p.8), Jarrick attributes views to earlier knowledge historians (citing Secord and Östling, note 12) which I do not recognize in the works cited. In English we call this setting up an ‘Aunt Sally’ in order to throw stones at it. The author needs to quote passages from these texts in order to substantiate his criticisms.

3.    Opposition to prevailing views sometimes leads the author to exaggerate the converse, e.g. to claim (not argue) that contemporary knowledge historians (not named) refuse to distinguish research from the communication of research’ (p.6).

4.    The central argument is sometimes obscured by digression, e.g. the case of Kuhn and ‘intra-scientific logic’ (p.9). It might be said that the article aims at too many targets, hence my reference to ‘several articles’ at the beginning of my comment.

5.    The mere listing of cases that may deserve study in J’s ‘programme’ would be helpful in a seminar presentation but is too raw for a published article. Better to cite fewer examples and say something about each.

6.    The presentation is clear but sometimes verbose and could be pruned (e.g., no need to write ‘and so on, and so forth’ (p.12). The quotation on p.15 from one of J’s books could be cut, and also the quotations from Bacon, which, placed where they are, get in the way of the necessary explanation to the reader of what the article that follows is about.

7.    The English is almost always correct – though the ‘Yet, also’ on p.6 would be more idiomatically expressed as ‘But these activities also required…’

8.    A few small omissions. Note 36 should cite Edney’s work on Rendell, which I am sure the author knows. On the ‘generation effect’, p.15, it would be worth citing (once again) the famous remark on the topic by Max Planck on scientific progress from funeral to funeral. The discussion of the relation between local and universal knowledge would benefit from including Jan Golinski, Making Natural Knowledge.

Author Response

First of all, I am thankful to the reviewer for her/his thorough comments. I agree to most of them and will try to revise my article accordingly.

  1. I agree that I make too frequent use of Wootton, who's works I, though, find very useful. I'll tone down my references to him, and make more use of some other researchers, such as Rens Bod, especially since he in  his impressive "A World of Patterns" applies a broader  perspective on global knowledge history than Wootton, more in line with the reviewer's point that I confine myself too much to the history o science. However, one has to limit onesef, one way or another, and I think that my basic point could be made with the examples chosen.
  2. Good point. I'll check Secord and Östling once more, and either quote them or drop to the notion.
  3. I think I am correct here, at least when it comes to some of knowledge historians. But I will check once more and try not to exaggerate their views.
  4. Correct. This is an unnecessary digression. I'll drop it.
  5. To be seriously considered. I will give fewer example and say a little more about each.
  6. The self-quotation will be cut, but the quotations of Bacon deserve being there since knowledge researchers refere to Bacon over and over again, as a point of departure, however wiuhtou having read his "The New Organon" (which I have -- it becomes boring after i while).
  7. Agree.
  8. Good suggestions. I'll check them again.

Round 2

Reviewer 2 Report

I have now read the revised version of the article and find it much clearer, more coherent and more fully argued than the first version. I recommend the publication of the new version.
